# Targeting Epigenetic Modifiers of Tumor Plasticity and Cancer Stem Cell Behavior

**DOI:** 10.3390/cells11091403

**Published:** 2022-04-21

**Authors:** Vigneshwari Easwar Kumar, Roshni Nambiar, Cristabelle De Souza, Audrey Nguyen, Jeremy Chien, Kit S. Lam

**Affiliations:** 1Department of Biochemistry and Molecular Medicine, UC Davis Medical Center, Sacramento, CA 95817, USA; vigneshwarie@outlook.com (V.E.K.); roshninamz@gmail.com (R.N.); cdesouza@stanford.edu (C.D.S.); adfry@ucdavis.edu (A.N.); kslam@ucdavis.edu (K.S.L.); 2Department of Stem Cell Research and Regenerative Medicine, School of Medicine, Stanford University, Stanford, CA 94305, USA; 3Department of Obstetrics and Gynecology, UC Davis Medical Center, Sacramento, CA 95817, USA

**Keywords:** cancer stem cells, epigenetics, signaling, tumoral plasticity, inhibitors

## Abstract

Tumor heterogeneity poses one of the greatest challenges to a successful treatment of cancer. Tumor cell populations consist of different subpopulations that have distinct phenotypic and genotypic profiles. Such variability poses a challenge in successfully targeting all tumor subpopulations at the same time. Relapse after treatment has been previously explained using the cancer stem cell model and the clonal evolution model. Cancer stem cells are an important subpopulation of tumor cells that regulate tumor plasticity and determine therapeutic resistance. Tumor plasticity is controlled by genetic and epigenetic changes of crucial genes involved in cancer cell survival, growth and metastasis. Targeting epigenetic modulators associated with cancer stem cell survival can unlock a promising therapeutic approach in completely eradicating cancer. Here, we review various factors governing epigenetic dysregulation of cancer stem cells ranging from the role of epigenetic mediators such as histone and DNA methyltransferases, histone deacetylases, histone methyltransferases to various signaling pathways associated with cancer stem cell regulation. We also discuss current treatment regimens targeting these factors and other promising inhibitors in clinical trials.

## 1. Introduction

Genetic and epigenetic tumor heterogeneity play an integral role in regulating tumor development, progression, cancer cell survival and therapeutic response. While personalized therapeutics are an attractive option in precision cancer medicine, approaches are complicated by the existence of genetic and epigenetic variations found within the same tumor (intratumoral heterogeneity) [1] (see Figure 1). Moreover, clonal evolution and genetic diversity in cancer cells can be attributed to cellular heterogeneity in tumors leading to the natural selection of genetic and epigenetic profiles shaped by tumor microenvironment and cellular adaptations to new metastatic niches during the epithelial–mesenchymal (EMT) transition [2]. The genetic/mutational heterogeneity then manifests as phenotypic variations observed in tumors with alterations in classical signal transduction pathways. This genetic and resulting phenotypic heterogeneity ensures the selection of fitness traits in primary tumors and promotes adaptation in metastatic secondary tumor sites [3]. In addition to mutational heterogeneity, there are epigenetic alterations that play an integral role in modulating cancer cell behavior. Epigenetic changes play an important role in determining cell fate and distinct tumor lineages. Of the various tumor cell lineages present, a specific subset of quiescent tumor cells with enhanced chemotherapy resistance, radiotherapy resistance and metastatic potential characterized as cancer stem cells (CSCs) or tumor-initiating cells (TICs) are implicated in tumor relapse [3].

Here we review some common epigenetic regulatory pathways controlling tumor cell plasticity and cancer stem cell properties. We also focus on the EMT transition and the epigenetic factors controlling invasion and metastasis in aggressive tumor types. Finally, we discuss current therapeutics being developed and utilized to target epigenetic vulnerabilities in CSCs for the treatment of cancer. 

## 2. Cancer Stem Cells

Cancer stem cells (CSCs) are a subpopulation of tumor cells within bulk tumor tissue characterized by multipotency, enabling them to self-renew and differentiate into distinct tumor lineages [4]. Long-term self-renewal of CSCs allows proliferation of cells while maintaining an undifferentiated state, aiding indefinite persistence of quiescent CSCs in tumor tissues [5]. A CSC undergoes asymmetric division, producing two daughter cells with differential fate: one retains the stem cell identity, thus maintaining the self-renewal capacity, while the other becomes a specialized progenitor with capacity to produce mature tumor cells and populate the bulk of tumor tissue [6]. Accordingly, the genetic and epigenetic heterogeneity within the CSC subpopulation contributes to intratumoral heterogeneity, and the self-renewal capacity of the CSC subpopulation is a factor in driving tumor relapse [7]. With some exception, most of the CSCs have high resistance to chemo- and radio-therapies due to a range of factors, including a favorable tumor niche, overexpression of drug efflux pumps, pro-survival and anti-apoptotic signals, intracellular drug-inactivating enzymes and enhanced DNA repair [8]. Therefore, targeting the vulnerabilities in CSC is critical for sensitizing tumors to cancer therapeutics and preventing relapse and metastasis. 

## 3. Cancer Stem Cells and Clonal Evolution as the Source of Intratumor Heterogeneity

The heterogeneity in cancer has been described by two models: the hierarchical cancer stem cell model and the stochastic clonal evolution model (Figure 2) [9]. In the hierarchical CSC model, it is hypothesized that a pre-existing CSC undergoes self-renewal and differentiation to produce both CSC and the non-CSC tumor cells that form the bulk of the tumor tissue. The CSCs may acquire tumorigenicity through clonal evolution and subsequent differentiation into non-CSC tumor cells via epigenetic mechanisms producing phenotypic heterogeneity [10,11]. The stochastic clonal evolution model hypothesizes that some tumor cells may accumulate genetic mutations over extended periods, and these genetic mutations contribute to phenotypic heterogeneity [9]. Studies describing tumor heterogeneity by the CSC model [10,11], stochastic evolution model [12] and non-CSC dedifferentiation into tumorigenic CSCs [13,14] have shown that CSCs are more resistant to conventional therapeutics than their non-CSC counterparts. Improved understanding of these CSC models is required since the true nature of CSCs will influence strategies for therapeutics and help lay the foundation for understanding critical properties of tumoral plasticity. The CSC model has significant clinical implications making it critical to target cancer stem cells to ensure complete elimination of tumor cells.

## 4. Cancer Stem Cell Plasticity

Tumor cell plasticity is defined as the property of cancer cells to exist in a dynamic stem/non-stem cell phenotype, epithelial/mesenchymal transitions or quiescent/proliferating condition [15]. This property permits CSCs to switch phenotypic states in response to environmental signals and is governed by intrinsic factors such as transactivators of EMT [16] and extrinsic factors such as signals transduced by the tumor microenvironment (TME) [17]. Tumor cell plasticity is characterized by epigenetic dysregulation caused by altered signaling from the TME, as well as variations in the pathways of metabolic glycolysis, oxidative phosphorylation (OXPHOS) and cell cycle progression [18]. The epigenetic mechanisms responsible for tumor plasticity include, but are not limited to: chromatin remodeling, histone modifications, DNA methylations or indirect control through noncoding RNAs, and mutations of cell cycle regulators such as p16 [19] and p21 [20]. Cellular plasticity is a major contributor to intratumoral heterogeneity [2]. With so many potential phenotypes within a single tumor, targeted therapeutic interventions may prove effective against some, but not all subpopulations, leaving resistant variants to grow back as relapsed tumors [21]. For example, considering that conventional chemotherapeutics target actively proliferating cells, quiescent CSCs may escape the treatment. 

As described above, maintenance of cancer stem cell plasticity is essential to promote tumor cell survival. This maintenance of the stem cell state in tumor cells is governed by genomic rearrangements, DNA mutations, epigenetic modifications and microenvironmental cues [2]. As opposed to genetic changes, phenotypic variations through epigenetic switches are incorporated with each cell cycle and are thus responsive to immediate changes in the microenvironment [2]. It has been reported that during cancer initiation and growth, mutations in epigenetic regulators enhance tumor plasticity favoring adaptability and resistance to therapy thus significantly dictating cell fate [22,23]. Since most epigenetic dysregulation mechanisms are both heritable and reversible, therapies targeting epigenetic regulation are promising treatment strategies [24]. 

## 5. Epigenetic Regulation/Dysregulation in Cancer Stem Cells

Increasing evidence has shown that malignant transformation is not a consequence of genetic mutations alone, but also involves reversible epigenetic dysregulation [25]. Histones are proteins that are involved in the regulation of gene expression and silencing through interaction with DNA. For example, H3K9 trimethylation (addition of three methyl groups to lysine residue at position 9 on histone 3) together with deacetylation of histones H4 and H3 is associated with gene repression, while H4K8 (lysine residue at position 8 of histone 4) acetylation and H3K14 (lysine residue at position 14 of histone 3) acetylation along with H3S10 (serine residue at position 10 of histone 3) phosphorylation is associated with gene expression [26]. In cancer, several of these post-translational modifications can undergo dysregulation, driving intratumoral heterogeneity and leading to tumor subpopulations with novel epigenetic regulation. These epigenetic regulations are carried out mainly by histone writers, erasers and readers (Figure 3). 

### 5.1. Classification of Epigenetic Mediators: Writers, Erasers and Readers

#### 5.1.1. Writer Enzymes

Writers are enzymes that are involved in adding methyl or acetyl groups to specific amino acid residues of histones or methyl group to cytosine nucleotides of DNA [25]. Adding methyl groups to CpG islands present in DNA or histone tails helps prevent transcription by blocking transcription factors [25]. Transcription can be further prevented by the recruitment of proteins that bind to methylated sites, creating additional inaccessible binding sites on the chromatin [25]. 

DNA methyltransferases (DNMT) are writer enzymes that are involved in adding a methyl group to cytosine residues that are a part of CpG dinucleotides, forming 5-methylcytosine [27]. CpG dinucleotides are found in CpG islands in promoters [27]. Methylation of these regions in genes is associated with gene silencing [27]. DNMTs are divided into DNMT1 and de novo DNMT. De novo DNMTs are expressed largely during development and are important in the maintenance of methylation patterns in human embryonic stem cells, whereas during differentiation DNMT1 becomes highly expressed, with a reduction in de novo DNMTs [27]. DNMT1s are responsible for ensuring proper inheritance of epigenetic patterns during replication [27]. Several studies have shown the importance of DNMT1 in the maintenance of the CSC phenotype [28]. For example, a knockout of DNMT1 led to a reduction of cancer stem cell markers such as high expression levels of ALDH (aldehyde dehydrogenase), CD44+ and CD24+ in colon cancer cell lines [29]. 

Histone lysine methyltransferases (KMTs) are also a type of writer enzyme involved in the transfer of one, two or three methyl groups to specific lysine (K) positions at the tails of histones, which correspond to different biological responses [27]. Methylation at positions H3K4 (lysine residue at position 4 of histone 3), H3K35 (lysine residue at position 35 of histone 3) and H3K79 (lysine residue at position 79 of histone 3) is associated with gene expression, or open chromatin, while methylation at positions H3K27 (lysine residue at position 27 of histone 3), H3K9 (lysine residue at position 9 of histone 3) and H4K20 (lysine residue at position 20 of histone 4) is associated with gene repression [27]. A trimethylation pattern has been associated with transcriptionally active promoters, while monomethylation has been associated with active enhancers [30]. KMTs (lysine methyltransferases) can be subdivided based on the presence of the SET [Su(var)3–9 Enhancer-of-zeste and Trithorax] domain [30]. These enzymes are highly site-specific and are responsible for adding mono, di and tri methyl groups to histones [30]. A list of these enzymes is well detailed in C. Hon and Hawkins [30]. Several of the KMT enzymes are over-expressed in multiple cancers [27]. Enhancer of Zeste-Homolog 2 (EZH2), Enhancer of Zeste-Homolog 3 (EZH3) and H3K27(lysine residue at position 27 of histone 3) KMTs were found at elevated levels in glioma, breast and leukemia CSCs, and are credited with maintaining their quiescent state [31].

#### 5.1.2. Eraser Enzymes

Erasers remove epigenetic modifications from DNA and histones. Histone demethylases remove methyl groups from histones; KDMs (histone lysine demethylases) remove methyl groups from lysine specifically. Lysine-specific demethylases are categorized into two groups: the FAD (flavin adenine dinucleotide)-dependent KDMs, KDM1A and KDM1B can remove mono and dimethyl groups but not trimethyl groups [27]. The Jumonji-C domain-containing histone demethylases, the second group of KDMs, can remove trimethyl groups from histone lysine residues, as well as di and monomethyl groups [27]. These two KDM categories are further subdivided into KDM2, KDM3, KDM4, KDM5 and KDM6, with each demethylating a specific lysine residue [27]. In embryonic stem cells, KDM1A (also known as lysine specific demethylase 1 (LSD1)) expression levels are typically elevated, but are reduced during differentiation [32]. LSD1 was also reported to maintain the balance between H3K4 di/tri methylation and H3K27 trimethylation marks (bivalent heterochromatin) at regulatory regions of several developmental genes, thus suppressing differentiation-associated genes and maintaining pluripotency in embryonic stem cells [33]. LSD1 has been shown to be an important regulator in maintaining the CSC population in treatment-resistant breast cancer cell lines. Pretreatment with LSD1 inhibitors improved treatment sensitivity to doxorubicin treatment [34]. Increased levels of KDM5A and KDM5B have been associated with chemoresistance in cancer and appear to contribute to an increase in cancer cell proliferation [35]. High levels of KDM5B are associated with the repression of tumor suppressor genes and apoptosis-related genes [36]. DICER, an enzyme involved in processing of microRNAs that are regulating the EMT pathway, showed a decrease in expression in breast cancer cell lines after hypoxic exposure, even though KDM6A and KDM6B were enriched in the promoter region, highlighting the importance of oxygen to carry out demethylating activity of KDM6A and KDM6B [37]. KDM6A was inactivated by a hypoxic environment in myoblast cell line C2C12, resulting in sustained H3K27me3, preventing myogenic differentiation [38]. 

Histone deacetylases (HDACs) are another group of eraser enzymes that remove acetyl groups from histones, which leads to chromatin compaction. As a result, DNA becomes inaccessible to transcription factors for gene expression. These enzymes cannot bind to DNA directly and require repressor complexes to facilitate DNA binding, such as NuRD (nucleosome remodeling and deacetylase complex), CoREST (co-repressor for element-1-silencing transcription factor) and KDM [27]. Knockdown experiments reveal that HDAC7 epigenetically modifies transcription start sites and super-enhancers of oncogenes such as *C-MYC, CD44* and *BMI-1* and reduces the expression of CD49f in breast cancer stem cells [39]. A knockdown of HDAC7 led to a reduction of sphere-forming ability and in vivo tumor growth of classic CSC phenotypes, and HDAC7 levels of expression in CSCs were also found to be higher than in non-stem cancer cells [40]. Being downstream of class 1 and class 2 HDACs, HDAC7 could be important in a targeting strategy for eliminating breast and ovarian CSCs [39]. Class 3 HDACs, also known as sirtuins, are dependent on NAD+ (nicotinamide adenine dinucleotide) for their activity thus playing a role in the metabolic regulation of the cell [27]. Class 4 HDACs include HDAC11, which has also been correlated with increased expression in lung cancer stem cell lines and regulates the expression of *SOX2* [(sex determining region Y)-box 2], an important transcription factor to maintain CSC self-renewal [41].

DNA demethylation proteins are a third type of eraser involved in the oxidation of 5-methylcytosine (5mC), present in CG dinucleotides, into 5 hydroxymethylcytosine [27]. TET (Ten Eleven Translocation) demethylating proteins fall under this group of epigenetic regulators. TET1, TET2 and TET3 play pivotal roles in the development and maintenance of the stem cell phenotype [27]. TET2 was found to be involved in the regulation of genes associated with self-renewal and differentiation in hematopoietic stem cells [42], and mutations in TET2 have been associated with hematological malignancies [43]. 

#### 5.1.3. Reader Enzymes

Readers are proteins that recognize histone and DNA modifications and can sense chromatin conformation. Bromodomain belongs to bromodomain and extra-terminal domain (BET) family members and recognizes acetylated lysine [44]. BRD4 (bromodomain-containing protein 4), which belongs to the BET family, binds to an acetylated promoter, allowing transcription of target genes [44]. *C-MYC* expression is regulated by BRD4 binding to the promoter and enhancer [22]. *C-MYC* was targeted using a BRD4 inhibitor, JQ1, in medulloblastoma. Inhibition of BRD4 was found to downregulate transcription of *C-MYC*, which led to a further reduction in cell growth as well as cell cycle arrest at the G1 phase. Stem cell-associated pathways were found to be downregulated upon treatment with JQ1. NANOG, Nestin and SOX2 classical stem cell markers were downregulated, while MAP2 (microtubule associated protein 2), a differentiation marker for neurons, was found to be upregulated [45]. *C-MYC* was validated as the target of BRD4 using luciferase-expressing plasmids controlled by *C-MYC*, studied in Daoy cells. This work showed a decrease in luciferase activity in the presence of JQ1 [45]. In another study, BRD4 was found to be associated with the N-termini of TWIST and *WNT5A* gene expression which are essential regulators of the EMT pathway [46].

## 6. Other Regulators of Cancer Stem Cells

### 6.1. Long Non-Coding RNA

A large part of the human genome contains DNA that does not code for any proteins and was long assumed to be “*junk DNA*”. Over the years, research has uncovered that although a large portion of transcribed RNA does not code for proteins, it serves a vital function in the regulation of genes [47]. Many of these regulatory segments fall into the category of long non-coding RNAs (lncRNA), which are greater than 200 nucleotides in length. LncRNA can be found in intergenic regions, transcribed from introns, sense RNA or antisense RNA [48]. LncRNAs can act as decoys, guides and signaling molecules. Decoys can cause gene repression by blocking the binding of proteins to RNA, inhibiting transcription [47,49]. Alternately, guides can facilitate gene expression by helping transcription factors or multiprotein complexes such as PRC (polycomb repression complex) to bind to target genes [47]. LncRNAs can also play integral roles in the regulation of CSCs. In the case of liver cancer stem cells, increased expression of *lncTCF7* led to the activation of the Wnt pathway through the recruitment of SWI/SNF to the promoter of *TCF* [50]. In embryonic stem cells, lncRNAs were found to be important targets of the transcription factors *OCT4, SOX2, C-MYC, KLF4* (Krueppel-like factor 4) and NANOG, which are involved in maintaining pluripotency [51]. Knocking down *lncTCF7* expression caused a reduction in the expression of *OCT4*, *SOX2*, *C-MYC*, *KLF4* and *NANOG*, along with reduced sphere-forming ability of liver cancer stem cells [50]. High XIST (LncRNA X inactive specific transcript) expression, a long non-coding RNA, was correlated with low miR-200c expression levels, and was identified as a potential target to eradicate bladder cancer stem cells [52].

### 6.2. ATP-Dependent Chromatin-Remodeling Complexes

Dynamic modification of chromatin is performed by ATP-dependent chromatin-remodeling complexes that utilize ATP to slide (translocate or move the histone along the DNA), evict or replace nucleosomes, thereby affecting gene expression [53]. Depending on the catalytic unit and associated subunits, the four classes of chromatin remodelers are SWI/SNF, CHD, ISWI and INO80. These four classes of remodelers possess an ATPase domain which is conserved across eukaryotes and function as multi-subunit complexes in association with tissue-specific subunits [54,55]. Epigenetic modification occurs through the binding of these complexes to specific chromatin domains such as bromo, chromo and SANT domains [56]. 

## 7. Hypoxia on Cancer Stem Cells

The hypoxic tumor microenvironment in solid tumors has been reported to play a role in CSC maintenance and reprogramming of non-CSCs to CSCs [57,58]. Hypoxia-inducible factor 1α (HIF1α) and factor 2α (HIF2α) are examples of primary mediators of hypoxia reported to be co-opted by CSCs to maintain stemness as well as evade hypoxia by enabling EMT. HIF1α-activated Notch signaling pathway was reported to be a critical regulator of CSC maintenance in hypoxic glioblastoma stem cells (GSC) [59]. In in vitro models of U251 cells, HIF1α mRNA and protein levels were found to be elevated along with the STAT3 and PI3K/Akt pathways in GSC. Inhibition of these pathways also partly reduced the hypoxia-mediated activation of Notch pathway [59]. HIF2α-induced release of VEGF was shown to promote angiogenesis in hypoxic TMEs in glioblastomas [57]. In in vitro stem cell models derived from patient biopsies and in vivo xenograft models, HIF2α was reported to be preferentially expressed in GSC vs non-GSC. Inhibition of HIF2α also reduced self-renewal and proliferation of GSC in vitro and decreased tumor initiation potential of GSC in vivo [57]. 

Hypoxia also regulates tumoral plasticity and EMT transitions in CSCs. Hypoxia mediated EMT plays an important role in inducing metastatic cells with CSC properties [60,61]. Hypoxia induced Jagged2 promoted metastasis in breast cancer in vitro as well as acquisition of stem cell phenotype by activating Notch signaling [60]. In in vitro and xenograft models of gastric CSCs, HIF1α induced EMT through the SNAIL pathway [61]. CSC maintenance in hypoxic conditions is also aided by glycolysis. Inhibition of glycolysis pathways has been reported to suppress CSC maintenance [62]. Pyruvate dehydrogenase kinase 1 (PDK1) is a glycolytic enzyme involved in the tricarboxylic acid cycle. PDK, a downstream target of HIF1α, is elevated in breast CSCs through the lncRNA H19/miRNA let-7/HIF1α signaling axis. siRNA mediated knockdown of PDK1 resulted in loss of colony formation and related stemness features in breast CSCs both in vitro and in vivo [62]. Given that hypoxia influences cell signaling pathways affecting CSC plasticity and maintenance, targeting HIFs and other signaling pathway factors influenced by hypoxia may be a relevant strategy to drive the loss of stem-like state in CSCs. 

## 8. Cell Signaling Pathways Regulating Cancer Stem Cells

Various cell signaling pathways in normal stem cells that regulate cell survival, growth, differentiation and self-renewal are found to be mutated in CSCs. Commonly implicated signaling pathways include Wnt, Notch, hedgehog, NF-κB and PI3K/PTEN [3]. Regulation through these pathways is not linear, but brought about by cross-talk between the pathways with shared subunits such as Gli and GSK-3β, [63] which aid the maintenance of CSC populations [63]. In this manner, genetic and epigenetic modifications of subunits participating in these signaling pathways can influence multiple other pathways and affect CSC maintenance. Therefore, identification of the main effectors of epigenetic regulation of signaling pathways is increasingly viewed as a potential area for therapeutic research.

### 8.1. Wnt Signaling Pathways

The canonical wingless and integration site growth factor (Wnt/β-catenin) signaling pathway is recognized as a crucial modifier of tumor cell plasticity [64]. Gene activation through the canonical Wnt signaling pathway is mediated by the transcription factor β-catenin. In the absence of Wnt ligand, the β-catenin destruction complex consisting of adenomatous polyposis coli (APC), Axin, Dishevelled-1 (DVL-1), glycogen synthase kinase 3 β (GSK-3β) and casein kinase 1 (CK1) prevents β-catenin translocation to the nucleus [65]. GSK-3β phosphorylates β-catenin for ubiquitination and proteasomal degradation. However, upon binding of the Wnt ligand to Wnt receptor Frizzled and Wnt co-receptor Low density lipoprotein receptor-related protein (LDR5/6), DVL is activated and causes disassembly of the degradation complex [66]. Dephosphorylated β-catenin, now free, translocates into the nucleus and associates with transcription factors T-cell factor/lymphoid enhancer (TCF/LEF), inducing expression of Wnt target genes such as *CCND1* and *MYC* [67]. Epigenetic regulation of Wnt signaling is reported in mice models of glioma, wherein activation of Wnt/TCF4 signaling causes overexpression of histone demethylase KDM4C, which removes H3K9me3 from Wnt target genes, promoting cell proliferation and tumorigenicity [68]. Meanwhile, DNA promoter methylation of secreted Frizzled receptor protein 1 (SFRP1) in hepatoblastoma was also found to correlate with mutated β-catenin [69]. Moreover, a loss of methylation at H3R8me2(arginine residue at position 8 of histone 3) and H4R3me2 (arginine residue at position 3 of histone 4) promoter regions of *AXIN2* and Wnt Inhibitory Factor 1 (*WIF1)* caused by the inhibition of protein arginine methyltransferase 5 (PMRT5) promoted lymphoma cell survival in patient-derived NHL cell lines [70]. 

### 8.2. Notch Signaling Pathway 

Notch signaling is an evolutionarily conserved pathway regulating differentiation and self-renewal in stem cells [71]. It is a contact-based signaling pathway wherein binding of ligands such as Jagged 1/2 or Delta1–4 causes the release of Notch intracellular cytoplasmic domain (NICD) into the cytoplasm through γ-secretase-induced cleavage of the transmembrane Notch receptor [72]. NICD then translocates to the nucleus which is followed by the activation of Recombination Signal Binding Protein 1 for J- Kappa (RBPJ- κ) for expression of *MYC* and *HES1* Notch genes [73]. Depending on the tumor microenvironment, the Notch pathway is involved in oncogenic or tumor suppressive roles in a wide range of tumors. It is reported that aberrant Notch signaling promotes CSC self-renewal and metastasis in ovarian, breast and HCC (hepatocellular carcinoma) CSCs [71]. In clear cell renal cell carcinoma (CCRCC), overexpressed Jagged1 was associated with DNA demethylation of H3K4me1-associated enhancer regions [74]. Epigenetic profiling of osteosarcoma cells identified leukemia inhibitory factor (LIF) as an essential factor controlled by super enhancers. LIF activation of NOTCH1 signaling by H3K27me3 demethylation through the Ubiquitously Transcribed Tetratricopeptide Repeat on chromosome X (UTX) histone demethylase correlated with “stemness” related genes, sphere formation, self-renewal and metastasis in osteosarcoma [75]. In cutaneous T cell lymphoma (CTCL), the binding of the epigenetic reader bromodomain-containing protein 4 (BRD4) to promoter and enhancer regions of CD4^+^ T cells of CTCL patients resulted in tumorigenesis through increased expression of *NOTCH1* and *RBPJ* genes [76]. 

### 8.3. Hedgehog Signaling Pathway

In CSCs, the hedgehog (Hh) signaling pathway is implicated in driving tumor growth, development and regulation of tumors after therapeutic intervention [77,78]. The hedgehog signaling pathway is mediated through (Glioma associated oncogene homologue) Gli proteins, which, in the absence of Hh signals are sequestered by Suppressor of Fused (SUFU) and Kinesin family member 7 (Kif7) [79]. However, upon the binding of Sonic hedgehog (Shh) ligand to the Patched receptor (PTCH1), Smoothened (SMO) is no longer suppressed by PTCH1 and becomes activated. Once activated, SMO causes the release of the sequestered Gli proteins, which translocate to the nucleus and regulate cell growth, proliferation and differentiation [71]. In patient-derived colorectal cancer-initiating cells, Indian hedgehog *(IHH)* gene was found to exist in a bivalent state with both the activating H3K4me3 and repressive H3K27me3 histone marks. The presence of such a bivalent state in the *IHH* gene aided stem cell maintenance of the colorectal cancer-initiating cells. Subsequent disruption of bivalency through the inhibition of EZH2 resulted in decreased self-renewal [77]. In gastric adenocarcinoma, significant DNA methylation of Shh transcription factors *CDX1/2* and *KLF5* correlated with decreased expression of *CDX1* and *KLF5* and an increased expression of *CDX2*. The increased expression of *CDX2* was associated with formation of metastatic lymph nodes in patients [78]. Similarly, in bladder cancer, it was reported that DNA hypermethylation of the CpG shore of the *SHH* gene caused the loss of the *SHH* gene and resulted in invasive urothelial carcinoma [80]. 

## 9. Epithelial–Mesenchymal Transition Is a Critical Regulator of the CSC Phenotype

Epithelial–mesenchymal transition (EMT) is a reversible transition process wherein cells in the primary tumor site acquire mesenchymal attributes to gain motility, and upon the invasion of a secondary tumor site ultimately revert to an epithelial state through a mesenchymal–epithelial transition (MET) [81]. In the primary tumor, cells are maintained in the epithelial state through epigenetic and transcriptional control of EMT genes [82]. Signals in the tumor microenvironment (TGFβ, WNT etc.) contribute to alterations in the expression of EMT genes, and can cause tumor cells at the margin to undergo the epithelial–mesenchymal transition, causing them to lose cell adhesion and polarity, dislocate from the primary tumor, gain cellular motility, survive in circulation through the body and infiltrate secondary organs [83]. E-cadherin is an adherens junction protein encoded by the *CDH1* gene that maintains the stemness of cells in the epithelial state. E-cadherin downregulation is a key step for initiating EMT and causing the detachment of cells from the bulk tumor [84]. E-cadherin expression is regulated by transcription factors Snail, Twist and Zinc finger E-box binding homeobox 1 (Zeb1/Zeb2) after activation of EMT [23]. Histone modifications that remove repressive H3K27me3 from promoter genes such as *ZEB1*, which are in the bivalent state (simultaneous methylations at different histone sites H3K27me3 and H3K4me3) also lead to alterations in EMT [82]. Such epigenetic alterations of transcription factors are mediated through chromatin remodeling complexes such as NURD and miRNA silencing, affecting the *ZEB1* promoter gene. ZEB1 represses epithelial genes such as *CDH1* that code for E-cadherin through epigenetic regulation and maintains the epithelial state of cells [85]. In NSCLC models, ZEB1 was reported to physically recruit the CHD4/NuRD complex to promoters and induce repression of miR-200c/141 and *TBC1D2b*, which suppress invasion and metastasis. This results in activation of Rab22, thus promoting metastasis through internalization and degradation of E-cadherin [85]. Invasive and migration properties of *ZEB1* are suppressed by miR-639. Repression of miR-639 by DNMT3A-mediated hypermethylation of its promoter gene was found to promote tumorigenesis in liver cancer mouse xenograft models [86]. Similarly, transcription factor Forkhead Box protein A2 (FOXA2) binding to the *CDH1* promoter caused a high expression of E-cadherin and reduced cancer cell migration in oral cancer [87]. Direct regulation of E-cadherin is also brought about by regulation of the *CDH1* gene. For example, in cancer cell line models of NSCLC, transcription factor Six2′s overexpression caused promoter methylation of *CDH1*, inhibiting E-cadherin expression and enhancing stemness and chemosensitivity [88]. Another epigenetic regulator, KDM7B, activates gene expression of EMT genes *SNAI1* and *VIM* by erasing repressive histone markers H3K9me1/2, H3K27me2 and H4K20me19 [89,90]. In HCC, it was reported that FIP-200-dependent autophagy caused PHF8-mediated repression of E-cadherin by *SNAI1* upregulation, which promoted EMT by causing degradation of E-cadherin [91]. 

Signaling pathways such as TGFβ are also known to affect epigenetic regulation of EMT genes [92,93]. In murine HCC models, TGFβ-mediated *SNAI2* upregulation resulted in an upregulation of H3K9 methylation and downregulation in H3K4 and H3K56 acetylation in E-cadherin promoter regions [92]. Additionally, this *SNAI2* upregulation also interacted with G9a and HDACs to suppress E-cadherin transcription, resulting in migration and invasion [92]. In another study, it was reported that in response to the exogenous TGFβ activation, PMRT5-MEP50 complex affected transcriptional activation by methylating H3R2me1 and H3K4, and repressed transcriptional activation through H4R3me2s demethylation of EMT genes and invasion pathways in lung and breast cancer cells [93]. Downstream targets of TGFβ signaling are also implicated in EMT. Epigenetic regulation of *RASSF10* (Ras-association domain family), a target of TGFβ, was reported to inhibit EMT by interacting with and stabilizing the *ASPP2* tumor suppressor gene [92]. It was reported that depletion of RASSF10 caused SMAD2 phosphorylation and promoted tumor invasion [94]. 

The epigenetic regulator Polycomb Repressive Complex 2 (PRC2) is found to be altered in GBM. Recently, the miR-490-3p/TGIF2/TGFBR1 axis was reported to affect EMT in GBM cell lines, wherein the EZH2 unit of PRC2 was found to regulate this miRNA and *CHRM2* host gene directly [95]. Another mechanism of EMT regulation is through the chromatin-modifying enzyme lysine-specific demethylase 1 (LSD1), which is overexpressed in colorectal cancers (CRCs). In an in vitro study of CRC cell lines, it was reported that LSD1-mediated AKT activity promoted EMT in cells containing the PIK3CA mutations [96]. Similarly, in another study on CRC cell lines, LSD1-mediated loss of H3K9me2 in the *TSPAN8* promoter region reportedly caused *TSPAN8* overexpression and promotion of EMT [97]. 

## 10. Integrins Alter Cancer Stem Cells Behavior

Integrins are heterodimeric cell surface receptors that can sense extracellular cues and signal intracellular processes governing proliferation, differentiation, adhesion and migration [98]. They have an extracellular portion, a transmembrane portion and a cytosolic tail [98]. Ligand binding induces a change in conformation of the integrin, leading to the recruitment of a cytoskeletal protein, talin, to the cytosolic portion of the β subunit [99], which activates focal adhesion kinases (FAK) and SRC family kinases [100]. Growth factors present in the ECM (extracellular matrix) require integrins to interact with their respective ligands as cofactors, in order to carry out their own signaling activities [100]. Unligated αVβ3 integrins can complex with KRAS to promote NFκB (nuclear factor kappa-light-chain-enhancer of activated B cells) activity, which has implications in cancer stem cell survival via self-renewal, proliferation and EMT (epithelial–mesenchymal) transition [101]. Integrins form dimers in various combinations of 18α and 8β subunits to form twenty-four sets of heterodimers [98]. Overexpression of integrins in cancer has been reported and various peptidic ligands against integrins have also been developed for targeted therapy and imaging [102,103].

αVβ3 integrins have been reported to contribute to resistance against receptor tyrosine kinase inhibitors [104]. An increase in ALDH (aldehyde dehydrogenase) activity, a marker for cancer progenitors, was observed in erlotinib-resistant lung cancer cells that displayed integrin β3+ compared to those cells that did not express integrin β3+, suggesting a contributing role of β3+ to the cancer stem cell phenotype [104]. In the same study, an increase in the expression of αVβ3 integrins was found in erlotinib-resistant lung tumor biopsies [104]. In another study, integrin α6 was highly expressed in glioblastoma cells that have a high expression of CD133, a cancer stem cell marker for glioblastoma [105]. α6(CD49f) has been used as a biomarker for identifying adult stem cells [106]. Interestingly, in the case of pluripotent stem cells, α6 expression is important to prevent FAK activation through β1 and to promote expression of genes associated with pluripotency and self-renewal [106,107]. 

Yes-associated protein (YAP) and transcriptional co-activator with PDZ-binding motif (TAZ) are transcriptional coactivators and have high expression levels in many cancers [108]. SRC activation was correlated with increased YAP/TAZ transcriptional activity [108]. A splice variant of α6 cytoplasmic domain (α6Bβ1) was found to activate TAZ, leading to transcription of genes associated with self-renewal [99], as well as the transcription of LMα5, which leads to the production of laminin (LM511) matrix [99,109]. The niche microenvironment surrounding cells has important functions in maintaining their stem cell properties [109]. LM511 has been shown to contribute to maintaining the stem cell phenotype upon interaction with integrins [109,110]. Autocrine VEGF signaling was found to control the α6B variant, but not α6A, and promote the breast cancer stem cell phenotype [99,111]. Overexpression of the α3 integrin in cancer cells that had acquired resistance was found to be associated with restrained metastasis through the suppression of Rho GTPase activity through Abl kinases in the Hippo tumor suppressor signaling pathway in prostate cancer [112]. However, in the case of glioblastoma, α3 was found to correlate with invasion and metastasis through ERK1/2 pathway [113].

Integrins can also mediate cell signaling events that regulate stem cell proliferation. For example, epithelial stem cells can differentiate into hair follicles and integrins β1, β4 and α6 are used to identify epidermal stem cells [114]. Other cells that support stem cells in their niche can secrete certain ligands that help to signal stem cells to remain in their quiescent state, as in the case of intestinal stem cells, where pericryptal fibroblasts secrete hedgehog ligands to inhibit proliferation [114]. Periostin plays an important role in creating a niche for metastatic cancer stem cells [115] and signaling through integrins is required for successful homing to the distant site [99]. Therefore, the survival of cancer stem cells is a result of extracellular signaling events in niches that help them maintain their quiescent state or promote their proliferative state. 

## 11. Targeting Vulnerabilities in Cancer Stem Cells

Key pathways in the maintenance of CSCs are regulated by epigenetic mechanisms [23]. Therefore, “epigenetic reconditioning”, or inducing differentiation of CSCs via drugs that modify epigenetic changes, such as hypomethylating agents, is one of the approaches utilized for targeting CSCs [116]. Since tumoral plasticity conferred by epigenetic modulators is reversible, it is an attractive approach for therapeutic interventions [117]. Epigenetic inhibitors targeting DNA methyltransferases, histone deacetylases, non-coding RNA and chromatin remodelers are able to reverse or inhibit epigenetic regulation of tumor cells and are currently being developed and investigated in different clinical phases [118,119,120].

### 11.1. Histone Methyltransferase Inhibitors

The EZH2 unit of the PRC2 complex is responsible for the repression of tumor suppressor genes and is frequently overexpressed in cancers, therefore EZH2 is currently being investigated in cancer therapy research [121]. 3-Deazaneplanocin A (DZNep) is one of the most studied EZH2 inhibitors. It indirectly affects EZH2 expression by inhibiting S-adenosylhomocysteine (SAH)-hydrolase, which causes an upregulation of SAH, leading to degradation of PRC2 and subsequent EZH2 inhibition [121]. Another EZH2 inhibitor, tazemetostat (TAZVERIK) was recently FDA approved for treatment of patients with epithelioid sarcoma and is undergoing trials for diffuse large B-cell lymphoma and mesothelioma as well [122]. Tazemetostat (EPZ6438) at a dose of 800 mg twice daily showed promising results in a phase I study that included relapsed or refractory non-Hodgkin’s lymphoma patients and solid tumor patients possessing the EZH2-activating mutations, and with patients that displayed a wild type tumor phenotype [123]. In the subsequent, ongoing phase II clinical trial by Epizyme, which is estimated to be completed by 2021 (NCT01897571) patients with epithelial sarcoma were grouped into two cohorts consisting of either treatment-naïve or relapsed patients [118]. Out of twenty-four naïve patients, the objective response rate (ORR) was 6% and disease control rate was 42%, as of the preliminary press release. Out of thirty-eight relapsed or refractory patients, ORR was 8% and the disease control rate was 16% [118]. A phase I study of patients with hematological malignancies and solid tumors involving GSK126 

(GSK2816126), which inhibits EZH2 directly by competing with S-adenosylmethionine (SAM), was terminated because 51% of the patients showed progressive disease, while only 34% showed stable disease (NCT02082977) [124]. A possible reason put forward for this failure was that GSK126 promoted differentiation of hematopoietic stem cells into myeloid-derived suppressor cells and reduced the anti-tumor activity of the inhibitor [125]. However, it was observed that a combination of GSK126 with gemcitabine/5-fluorouracil showed an increase in anti-tumor activity compared to the use of either agent as a monotherapy (Refer to Table 1) [125]. 

### 11.2. Histone Methylase Inhibitors

Overexpression of lysine-specific demethylase 1 (LSD1) is reported in aggressive forms of cancers [27]. LSD1 inhibitors are being investigated in solid tumors (metastatic breast cancer, NSCLC, Ewings sarcoma) post reported success in AML (acute myeloid leukemia) [126]. Tranylcypromine is an LSD1 inhibitor with known toxicities, but there are less toxic, chemically modified forms of tranylcypromine with enhanced specificity to LSD1, such as ORY-1001(RO7051790) and GSK2879552, that are undergoing investigation in clinical trials for solid tumors (Refer to Table 2) [34]. Recently, a dual class I HDAC and LSD-inhibitor Domatinostat (4SC-202) was reported to show cytotoxic and cytostatic effects on ATRT cells using in vitro 2D and 3D spheroid models [127]. 

### 11.3. DNA Methyltransferase Inhibitors

Therapy with azanucleosides such as 5-azacitidine and decitabine, which are analogues of cytidine, have been shown to activate silenced tumor suppressor genes in vitro [128]. Azanucleosides work either by being incorporated into DNA and RNA, triggering the DNA damage response pathway, or by directly inhibiting DNA methyltransferases [128] through trapping DNMTs, preventing them from further methylation activity during replication [129]. Earlier studies found azacitidine mostly being effective in the S phase of the cell cycle, and demonstrating cytotoxic effects [130]. However, at lower doses, the effect of azacitidine preferentially acts on DNMTs, thus activating silenced tumor suppressor genes and differentiation processes [131,132]. Currently, they are approved by the FDA for acute myeloid leukemia and myelodysplastic [128]. However, for solid tumors and other hematological cancers, there is no evidence to support azacitidine for use as a single agent [129]. Nonetheless, it may improve the response to immunotherapy [132]. The development of resistance to treatment in patients who achieved cancer remission has been reported, and the issue of resistance cannot be completely eliminated by using DNMT inhibitors alone [121,128]. A study found that in hematological malignancies such as MDS, the hematopoietic progenitors that did not respond to azacitidine had an upregulation of the *ITGA5* gene which codes for integrin α5 and is correlated with CSC quiescence [133]. Using an anti-α5 antibody with simultaneous azacitidine treatment to block α5 signaling, an improved hematopoiesis response was observed [133]. 

Strategies proposed for successful long-term survival without relapse include priming cancer cells to cytotoxic therapy by treating them with decitabine before chemotherapy, as put forward in a study consisting of non-small cell lung carcinoma (NSCLC) patients with a history of prior chemotherapy [134]. The study used a combination of azacitidine and entinostat, a HDAC inhibitor. For treating solid tumors, a combination of Vidaza, the commercial name of azacitidine from Celgene, with Abraxane (nanoparticle albumin-bound paclitaxel) administered to patients with breast cancer and other advanced and metastatic solid tumors showed a 61.5% better response to 75 mg/m^2^ azacitidine and 100 mg/m^2^ of Abraxane, versus the control (NCT00748553). The ORR for a time frame of 1.5 years was 53.8%. In the case of decitabine, at lower doses, older patients suffering from AML were able to tolerate treatment [129]. Additionally, a comparison between decitabine and azacitidine showed a higher ORR (overall response rate) and PFS (progression free survival) to a five-day decitabine treatment regimen every four weeks, versus a seven-day azacitidine treatment regimen. However, there were no significant differences between OS and EFS (event free survival) (Refer to Table 3) [135].

Non-nucleosides have been also used as inhibitors due to the side effects of using azanucleosides [136] EGCG (epigallocatechin gallate) is a polyphenol found in green tea. EGCG increased the sensitivity of colon cancer cells in vitro to subsequent treatment with 5-fluorouracil [136]. However, in phase II clinical trials assessing the effects of EGCG as a maintenance therapy to prevent recurrence in ovarian cancer patients who had previously undergone chemotherapy, five of the sixteen women in the study did not have an ovarian cancer recurrence, but twelve women faced grade 1 adverse events [137].

Hydralazine, another non-nucleoside inhibitor against DNMTs, was used earlier along with valproic acid, an HDAC inhibitor, for sensitizing patients suffering from advanced-stage solid tumors [138]. The phase II single arm study noted that 80% of patients had clinical benefit in terms of partial response and stable disease (Refer to Table 3) [138]. The combination of hydralazine and valproic acid as antimetastatic agents in Ras-transformed 3T3 cells demonstrated growth inhibition in vitro and in vivo [139]. Overall, azanucleosides seem to be more effective than non-nucleosides.

### 11.4. Histone Deacetylase Inhibitors

HDAC inhibitors can be classified into selective, non-selective and multi-pharmacological HDAC inhibitors [140,141]. Vorinostat is a non-selective HDAC inhibitor which targets class 1 and class 2 HDACs and is used in the treatment of cutaneous T cell lymphoma (CTCL) [131]. In relapsed/refractory Hodgkin’s lymphoma, vorinostat administered as a single agent did not lead to any significant changes. One out of twenty-five patients who had previous chemotherapeutic treatment had a partial response and twelve of the patients had stable disease [140]. A meta-analysis of clinical trials consisting of three HDAC inhibitors (vorinostat, panobinostat and ricolinostat) concluded that panobinostat is the most effective of these three HDAC inhibitors, with an ORR (Overall Response Rate) of 0.64 in patients with relapsed/refractory multiple myeloma [142]. The authors of the study also concluded that patients who had previously been treated with lenalidomide can have an improved outcome upon treatment with HDAC inhibitors. An HDAC6 inhibitor, ricolinostat, had better clinical outcomes in combination with bortezomib and dexamethasone, rather than as a monotherapy for patients with multiple myeloma (Refer to Table 4) [143]. Romidepsin is a cyclic peptide that can inhibit HDAC1 and HDAC2 specifically [144]. Though HDAC inhibitors have had positive clinical outcomes in hematological malignancies, in solid tumors, such success has not been seen [145]. Curcumin has strong anti-oxidative and anti-inflammatory properties [146] and has been demonstrated to improve treatment against cancer stem cells when in combination with the FOLFOX chemotherapy regimen in patient-derived ex vivo colorectal cancer models [147]. Belinostat, a pan HDAC inhibitor, is approved by the FDA as a single agent treatment for relapsed refractory T-cell lymphoma (PTCL) [148]. 

### 11.5. BET Inhibitors

BRD4 of the BET family is an epigenetic reader that binds to acetylated lysine on histones and is reported to play a role in tumorigenesis in a variety of cancers [44]. BET inhibitors bind to bromodomains, blocking recognition of acetylated lysine on histones and inhibiting acetylation [27]. JQ1 (small molecule) and I-BET762 (synthetic mimic) are heavily utilized competitive inhibitors of BRD4 [149]. JQ1 has been reported to suppress the expression of ALDH(aldehyde dehydrogenase) by targeting the super enhancer regions of ALDH1A1 [150]. In vitro and in vivo SMARCA4/A2-deficient ovarian and lung cancer models showed BETi mediated anti-proliferative effects, showing that SMARCA4/A2 (SWI/SNF-related, matrix-associated, actin-dependent regulator of chromatin, subfamily a, member 4/2) may be used as biomarkers for BETi sensitivity [127]. There has also been an interest in proteolysis targeting chimera (PROTAC)-based BET protein inhibition, as ARV-825 has been reported to possess antileukemic activity alone or in combination with cytarabine in murine models [151]. BMS-986158 is a BET inhibitor currently investigated for treatment of pediatric cancers in Phase I clinical trials (NCT03936465). ZEN003694 is another bromodomain inhibitor currently being investigated in Phase II clinical trials with a PARP inhibitor drug talazoparib as a treatment for patients with Triple Negative Breast Cancer (TNBC) (Refer to Table 5).

### 11.6. Inhibitors against Signaling Pathways

Given that epigenetic regulation of signaling pathways also affects tumor plasticity, drugs affecting signaling pathways may also be used for conferring epigenetic control [70,119]. Notch pathway inhibitors usually inhibit γ-secretase that is responsible for cleaving the ICD of the Notch receptor. A γ-secretase inhibitor called nirogacestat (PF-3084014) has been investigated in twenty-four patients with desmoid tumors in phase I and II clinical trials, showing good tolerability and is currently undergoing phase III trials (NCT01981551). DAPT is another γ-secretase inhibitor reported to increase cisplatin sensitization by inhibiting proliferation, increasing apoptosis and reducing motility in osteosarcoma CSCs alone or in combination with cisplatin, which shows that a combination of a γ-secretase inhibitor with other treatment modalities can improve drug efficacy [119]. However, γ-secretase inhibitors alone are not effective, as seen in clinical trials of T cell acute lymphoblastic leukemia where a γ-secretase inhibitor showed low efficacy due to the presence of γ-secretase inhibitor-resistant cells with BRD4-dependent transcription; hence, combination treatments are required to target treatment-resistant subpopulations [22]. Approved hedgehog (Hh) signaling pathway inhibitors targeting the Smo subunits vismodegib and sonidegib have reduced efficacies in unselected patients, Smo-mutated patients and in patients showing initial treatment response due to acquired resistance and/or cross resistance to these SMO-inhibitors, thus making the use of single agent therapy unfeasible [152]. Glasdegib by Pfizer is an oral Hh inhibitor approved for AML, currently in phase II clinical trials for myelodysplastic leukemia and chronic myelomonocytic leukemia [153]. Other inhibitors of Hh pathways are drugs that prevent the binding of Gli transcription factors to DNA such as arsenic trioxide (ATO), which is in phase II clinical trials [154], and GANT61, which is under preclinical study [155]. Napabucasin (BBI608) is an example of a STAT3 inhibitor found to deplete cancer stem cells in NSCLC and prostate cancer, and is currently being investigated in phase III clinical trials for use in metastatic colorectal carcinoma (NCT02753127) [120]. Mutations in *ARID1A*, a component of the SWI/SNF chromatin remodeling complex, are recognized as biomarkers of single-agent ATR checkpoint inhibition, and represent a synthetic lethal approach [156]. Use of ATR inhibitors as single-agent treatments are being investigated in an ongoing phase II clinical trial involving patients with *ARID1A* loss in gynecological cancer treated with an ATR-inhibitor in combination with olaparib (NCT04065269) (Refer to Table 6). Tumor plasticity driven by epigenetic regulators in response to environmental cues and therapeutic interventions assists tumor evasion and relapse. Therefore, a recognition and understanding of epigenetic heterogeneity is essential to develop synthetic lethality-based treatment modalities, since targeting isolated pathway mechanisms would leave behind tumorigenic subpopulations. 

## 12. Conclusions

In this review we discuss several key players that dictate and modulate the behavior of cancer stem cells. Current cancer therapeutic strategies still face challenges especially due to quiescent stem cell populations that promote tumor relapse. The heterogeneous nature of cancer makes it difficult to find one treatment that can kill all cancer cells. The dynamic nature of these cells, resulting from the accumulation of mutations over time causing malfunction of epigenetic factors or by tumor microenvironment allowing CSC phenotype survival, has made it challenging to find a one size fits all solution. Research on epigenetic factors affecting CSC phenotype maintenance and survival has uncovered key epigenetic signatures and related signaling pathways that are linked to CSC formation and maintenance.

FDA (US Food and Drug Administration) approved drugs targeting epigenetic factors, currently in the market, are a handful. Drug development in this area is gaining momentum with many inhibitor analogs against HATs, HDACs, BETs, TETs, HMTs and DMTs being tested as monotherapy or in combination with existing therapies in clinical trials. Currently, most of the approved drugs are indicated for hematological malignancies that tend to relapse. As epigenetic signatures involved in CSCs survival can also be found in normal cells, stem cells and tissues, targeted therapies may become necessary for improved efficacy and reduced off target effects. We provide a comprehensive list of several factors governing cancer stem cell populations in heterogeneous tumors. We also discuss the interplay between epigenetic modifiers and stem cell survival. In addition, our review focusses on therapeutics that target these epigenetic modifiers in current in vitro, in vivo and clinical studies. We provide a detailed understanding of the underlying determinants of the stem cell niche. Finally, we aim to inform the scientific and clinical community of the various therapeutic options available in clinic and point towards exciting new avenues that warrant further research.

An emerging field of epigenetics that is becoming more relevant to CSC maintenance and differentiation is bivalent chromatin biology. Initial discovery that promoter regions of developmentally regulated genes in embryonic stem cells (ESCs) contain both transcriptionally repressive mark (H3K27me3) and active mark (H3K4me3) [157] fosters the idea that chromatin bivalency is closely linked to the pluripotent state of stem cells and that stable selection of either mark allows lineage specification [158]. Subsequent discoveries of these bivalent chromatin domains in cancer suggest that this epigenetic pattern may play a role in genes regulating CSC phenotype [159] and clonal selection upon drug treatment [160]. As such, combinations of epigenetic therapies that affect chromatin bivalency may be effective in altering CSC stem properties and may be best-suited to limit heterogeneity and plasticity of CSCs [161]. Future studies in regulation and targeting of chromatin bivalency in cancer may hold the key to epigenetic targeting of cancer cell plasticity, heterogeneity and multipotency.

## Figures and Tables

**Figure 1 cells-11-01403-f001:**
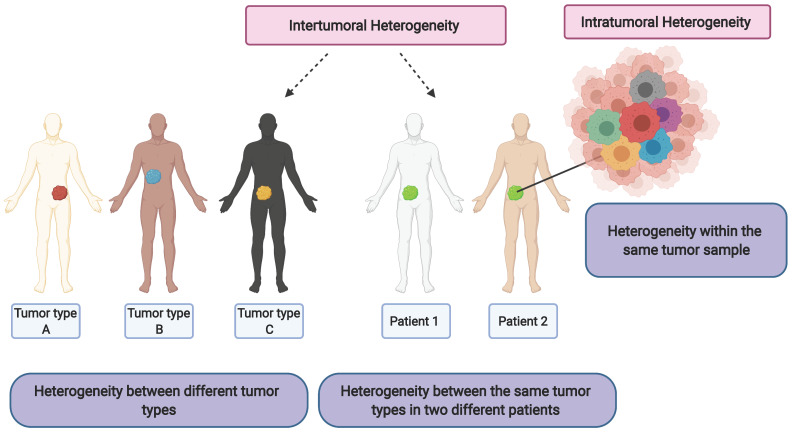
Conceptual diagram describing intertumoral heterogeneity and intratumoral heterogeneity. Created with BioRender.com (accessed date on 28 March 2022).

**Figure 2 cells-11-01403-f002:**
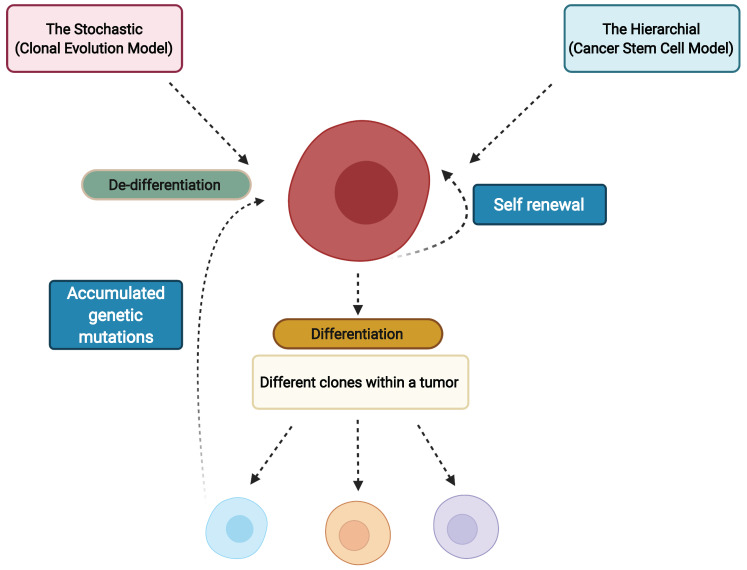
Schematic describing the two current models of cancer stem cells. Tumor heterogeneity is produced by clonal evolution and differentiation of CSCs into CSC and non-CSC tumor cells in the Hierarchical model while in the Stochastic model it is caused by the extended accumulation of genetic mutations in tumor cells. Created with BioRender.com.

**Figure 3 cells-11-01403-f003:**
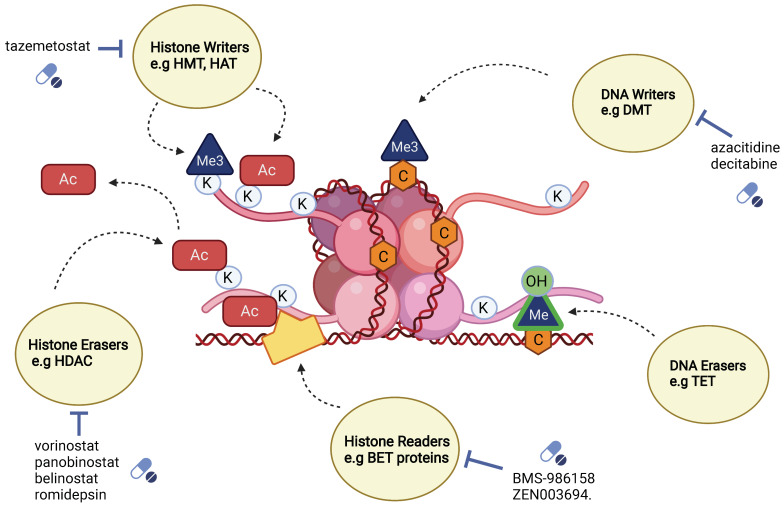
Conceptual diagram describing key epigenetic modifiers, their mode of action and their select inhibitors. Histone writers add methyl groups (Me3) or acetyl groups (Ac) can be inhibited by approved drug tazemetostat. DNA writers, such as DMT (DNA methyltransferases) remove methyl groups (Me3) and can be inhibited by approved drugs; azacitidine and decitabine. Histone erasers such as HDAC (histone deacetylases) remove acetyl groups (Ac) and can be inhibited by approved drugs; vorinostat, panobinostat, belinostat and romidepsin, while DNA erasers such as TET (ten-eleven translocation) oxidize 5-methyl cytosine into 5 hydroxymethyl cytosine. Histone readers such as the BET proteins can recognize acetylated lysine residues on histone tails, and they can be inhibited by BMS-986158 and ZEN003694. Created with BioRender.com.

**Table 1 cells-11-01403-t001:** Histone Methyltransferase Inhibitors in Clinical Trials for the treatment of cancer.

Study	Phase/ Randomization	Drug	Disease	Combination with
NCT01897571	Phase I/II	Tazemetostat	Epithelial sarcoma	Single agent
NCT02082977	Phase I	GSK126	Relapsed/refractory diffuse large B cell lymphoma, transformed follicular lymphoma, other non-Hodgkin’s lymphomas, solid tumors and multiple myeloma	Single agent

**Table 2 cells-11-01403-t002:** Histone Methylase Inhibitors in Clinical Trials for the treatment of cancer.

Study	Phase/ Randomization	Drug	Disease	Combination with
NCT02913443	Non-randomized	ORY-1001	ED SCLC *	Single agent
NCT02034123	Non-randomized	GSK2879552	Relapsed/refractory small cell lung carcinoma	Single agent

* Extensive stage disease small cell lung cancer.

**Table 3 cells-11-01403-t003:** DNA Methyltransferase Inhibitors in Clinical Trials for the treatment of cancer.

Study	Phase/ Randomization	Drug	Disease	Combination with
NCT00748553	Not available	Azacitidine	Breast cancer and metastatic solid tumors	Abraxane
NCT00404508	Non-randomized	Hydralazine	Refractory solid tumors	Magnesium valproate

**Table 4 cells-11-01403-t004:** Histone Deacetylase Inhibitors in Clinical Trials for the treatment of cancer.

Study	Phase/ Randomization	Drug	Disease	Combination with
NCT01583283	Not available	Ricolinonstat (ACY-1215)	Multiple myeloma	Lenalidomide and dexamethasone

**Table 5 cells-11-01403-t005:** BET Inhibitors in Clinical Trials for the treatment of cancer.

Study	Phase/ Randomization	Drug	Disease	Combination with
NCT03936465	Phase I	BMS-986158	Pediatric solid tumors, lymphomas, brain tumor	Monotherapy
NCT03901469	Phase II	ZEN003694	Triple Negative Breast Cancer	Talazoparib

**Table 6 cells-11-01403-t006:** Inhibitors against cell signaling pathways in Clinical Trials for the treatment of cancer.

Study	Phase/ Randomization	Drug	Disease	Combination with
NCT01981551	Phase I/II	Nirogacestat	Desmoid tumors	Monotherapy
NCT02753127	Phase III	Napabucasin (BBI-608)	Metastatic colorectal cancer	5-Fluorouracil, Leucovorin, Irinotecan
NCT04065269	Phase II	AZD6738	Gynecological cancers	Olaparib

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
