# Peer review of "Targeting Epigenetic Modifiers of Tumor Plasticity and Cancer Stem Cell Behavior"

_cells, 2022, doi:10.3390/cells11091403_

Round 1

Reviewer 1 Report

The review “Targeting epigenetic modifiers of tumor plasticity and cancer stem cell behavior” by Kumar et al. is well written and the different sections give a broad overview of the main features of cancer stem cells, the mechanisms underlying their plasticity and the role exerted by dysregulated post-translational modifications in maintaining the stem population and inducing intra-tumoral heterogeneity. In addition, the main signaling pathways responsible for cancer stem cell survival and self-renewal, the mechanisms regulating the EMT process and the current chemotherapeutic strategies (targeting the modified epigenetic mechanisms) employed in clinical trials are widely reported.

Minor revisions:

In the text, write the figures or tables in the same form: for example, Figure 1 is written in normal font (line 32), Figure 2 is written in bold and italics (line 72), Figure 3 in italics (line 133) and so on.

Line 54: delete the comma after (CSCs)

Paragraph 5. Epigenetic Regulation/Dysregulation in Cancer Stem Cells.

Why are some terms (acetylation, deacetylation) written in italics, while other ones (trimethylation, phosphorylation) not?

Paragraph 8. Epithelial Mesenchymal Transition is a Critical Regulator of the CSC phenotype.

Lines 357-360: maybe you have to delete “and” after “tumor site”

“Epithelial mesenchymal transition (EMT) is a reversible transition process wherein cells in the primary tumor site acquire mesenchymal attributes to gain motility, and upon the invasion of a secondary tumor site and ultimately revert to an epithelial state through a mesenchymal epithelial transition (MET)”

Author Response

Thank you for your suggested revisions. We revised the manuscript accordingly.

Reviewer 2 Report

General comment:

Despite not having a great novelty on this topic, this review is well written and provides a reader-friendly compendium of the main epigenetic targets and studies to date.

Minnor comment:

Considering the importance of the tumor microenvironment, I suggest at least mentioning the importance of non-cellular components, such as hypoxia, that affect the maintenance of the stem properties of CSCs and their epigenetic plasticity.  

Author Response

Thank you for your suggestion. We added a section highlighting relevant studies on the effect of hypoxia on CSC and their epigenetic plasticity.